

# Automated mineralogy based on energy dispersive X-ray fluorescence microscopy (μ-EDXRF) applied to plutonic rock thin sections in comparison to Mineral Liberation Analyser

Wilhelm Nikonow[1], Dieter Rammlmair[1]

[1]Federal Institute for Geosciences and Natural Resources (BGR), Stilleweg 2, 30655 Hannover, Germany

*Correspondence to*: Wilhelm Nikonow (w.nikonow@t-online.de)

**Abstract.** Recent development in the application of energy dispersive X-ray fluorescence spectrometry mapping (μ-EDXRF) has opened new opportunities for fast geoscientific analyses. Acquiring spatially resolved spectral and chemical information non-destructively for large samples of up to 20 cm length provides valuable information for geoscientific interpretation. Using

supervised classification of the spectral information, mineral distribution maps can be obtained. In this work, thin sections of plutonic rocks are analyzed by μ-EDXRF and classified using the supervised classification algorithm Spectral Angle Mapper (SAM). Based on the mineral distribution maps, it is possible to obtain quantitative mineral information, i.e. to calculate the modal mineralogy, search and locate minerals of interest and perform image analysis. The results are compared to automated mineralogy obtained from the Mineral Liberation Analyser (MLA) of a Scanning Electron Microscope (SEM) and show good

accordance, revealing variation resulting mostly from the limit of spatial resolution of the μ-EDXRF instrument. Taking into account the little time needed for sample preparation and measurement, this method seems well suitable for fast sample overviews with valuable chemical, mineralogical and textural information, and additionally, enabling the researcher to make better and more targeted decisions for subsequent analyses.

**Keywords**: μ-EDXRF, ENVI, MLA, automated mineralogy, mineral classification, image analysis

## 1 Introduction

Energy dispersive X-ray fluorescence microscopy (μ-EDXRF) is a versatile technique commonly used in various fields like art (Keune et al., 2016), archeology (Kozak et al., 2016), biology (Figueroa et al., 2014), medicine (Wandzilak et al., 2015) and also with increasing extent in geosciences (Belissont et al., 2016; Croudace and Rothwell, 2015; Flude and Storey, 2016;

Gergely et al., 2016; Kéri et al., 2016; Lombi et al., 2011; Melcher et al., 2006; Poonoosamy et al., 2016; Rammlmair et al., 2001; Rammlmair et al., 2006; Redwan et al., 2016), mostly for visualisation and quantification of element distributions. The combination of spatial and spectral information for large samples of up to 20 cm length with almost no sample preparation opens many fields of applications. It provides quick textural and chemical overview with spatially resolved main and trace element information (Nikonow and Rammlmair, 2016) at relatively low cost and easy operability compared to e.g. a SEM.



Applying supervised spectral classification, the chemical data can be used to derive mineral maps for quantitative petrography and the modal mineralogy (Nikonow and Rammlmair, 2016), which is key information for e.g. rock classification (Streckeisen, 1976).

The development of automated mineralogy was mainly driven by the ore mineralogists recognizing the value of automated

phase recognition and of defining the liberation degree of valuable minerals. With the application of image analysis to optical microscopy a first step was done (Russ, 1990; Starkey and Samantaray, 1994). The limitation regarding separating ore minerals of similar reflectance was overcome by development in acquiring backscattered electron (BSE) images from the electron microprobe (Petruk, 1988; Petruk, 1986) and scanning electron microscopy (SEM), combined with an energy dispersive X-ray fluorescence analysis (EDXRF) based mapping tool, QEMSCAN (Gottlieb, 2008; Gottlieb et al., 2000; Grant et al., 1976).

To speed up the acquisition time, up to 4 EDXRF detectors were finally involved. Later, with Mineral Liberation Analyser (MLA) a new approach was chosen, based on BSE image grey values and centered analysis of each detected grain (Gu, 2003). Until today, various devices combining SEM and EDXRF are on the market.

Automated mineralogy based on SEM and EDXRF is commonly used in different fields, e.g. in the mining industry. It is applied in exploration ore characterization or mineral processing (Sandmann, 2015; Sutherland and Gottlieb, 1991) and

15 provides high resolution data, but requires preparation of polished or thin sections, coating, a skilled operator and several hours up to days for measurement.

For modal analysis and automated mineralogy three types of errors have to be taken into account (Solomon, 1963): First, the false classification of a mineral, which is dependent on the operator designed mineral database (operator error). Then the 2D effect, since 1D information is extrapolated to a 2D area (counting error). The third type of error is the 3D effect resulting from

20 extrapolation of the 2D sample surface to the 3D properties of the sample (sampling error). Being able to quantify these errors, the classification data becomes more reliable. Using µ-EDXRF and spectral classification, the reliability of the classification algorithm can be measured by the classification thresholds. Mapping the whole sample surface will eliminate the 1D extrapolation effect; only the extrapolation to the 3D volume of the sample will remain, but can be decreased by mapping series of rock slices or thin sections due to progress in measurement speed and partial automatization of it.

In this work we evaluate the utility of µ-EDXRF based spectral classification and image analysis of thin sections using hyperspectral software (ENVI) for automated mineralogy compared to SEM + MLA.

## 2 Material and methods

### 2.1 Samples

Three polished thin sections of plutonic rocks from the collection of the Federal Institute of Geosciences and Natural Resources

(BGR, Germany) were selected to be analyzed both with µ-EDXRF and SEM + MLA. As far as possible, samples with different properties regarding mineral content and grain size were selected. Prior to this work the thin sections were analyzed under a polarization microscope and classified as quartz diorite (sample 484), monzogranite (sample 342) and syenogranite



(424) with varying mineral content regarding the main minerals quartz, feldspars, biotite, hornblende and pyroxenes as well as trace minerals like magnetite, ilmenite, titanite, calcite, apatite, zircon and allanite. The thin section size is about 35 x 23 mm.

## 2.2 Energy dispersive X-ray fluorescence microscope (µ-EDXRF)

For µ-EDXRF data acquisition the M4 Tornado from Bruker (Bruker, 2010) was used. The X-ray radiation is generated by a tube with a Rh-target operating with a maximum power of 30 W. The polychromatic beam is focused by a polycapillary lens resulting in a spot size of 17 µm at 17.48 keV (Mo Kα). In this work the M4 Tornado is equipped with two silicon drift detectors (SDD) facing each other at 180° and 90° to the tube. The maximum tube excitation of 50 kV and 600 µA was chosen in order to measure trace elements like REE and Zircon. The thin sections were measured with a step size (i.e. pixel size) of

12 µm and a dwell time of 3 ms per pixel. The total resolution of the measurement is about 2200 x 1600 pixels. The measurement takes about 3 h for one detector. In order to eliminate the effect of diffraction, the samples were measured with both detectors separately and the minimum intensity for each pixel was calculated and used for the classification. For more details regarding diffraction elimination see Nikonow and Rammlmair (2016). The measurement data was saved into a data cube which contains a full spectrum for each measured pixel. With the M4 Tornado software these results can be presented as

element distribution maps with element intensities in false colors or grey scales. From these element maps regions of interest (ROI) can be selected and quantified chemically using the fundamental parameter approach.

## 2.3 Scanning Electron Microscope (SEM) and Mineral Liberation Analyzer (MLA)

For comparison and validation of the µ-EDXRD based mineral classification and analysis, the SEM FEI Quanta 650 F combined with MLA was used. For this work the XBSE mode of MLA was applied on the samples 424 and 342, where grains

in the BSE image are classified and separated according to their grey level. Then each separated grain is measured in the center with the X-ray detectors and classified chemically using a predefined mineral database. For the sample 484 the XBSE was not suitable, since the grey values of hornblende and biotite were too similar for a correct grain separation. This sample was measured in GXMAP mode, where the whole sample was mapped with a distance of 6 µm. The details of the SEM image acquisition are listed in Table 1. A detailed description of the functionality of MLA can be found elsewhere (Dobbe et al.,

2009; Fandrich et al., 2007; Gu, 2003).

## 2.4 Mineral classification of element distribution maps

For analysis of the element distribution maps obtained by M4 Tornado, ENVI 5.1 by Exelis (Exelis Visual Information Solutions, 2015) was used. The supervised classification algorithm Spectral Angle Mapper (SAM) (dos Reis Salles et al., 2016; Kruse et al., 1993) was applied on the 2D-data from M4 Tornado for the mineral classification. The classification algorithm





allocates a mineral name to each pixel in the element distribution map according to a prior defined database of mineral spectra (endmember collection).

To establish this database, thin sections were studied under a polarization microscope. Some minerals were also analyzed by Electron Microprobe Analysis (EMPA) (Table 2). Knowing the mineral name and its location on the thin section, the spectra

of the corresponding pixels of the EDXRF measurements were defined as mineral endmembers for the mineral database. Additionally, EDXRF spectra of selected areas like mineral grains can be selected to calculate a sum spectrum and quantify the element ratios for a quantitative chemical analysis using the Bruker fundamental parameter algorithm. Fig. 1 shows the spectrum of a K-feldspar and in Table 2 the quantification results are listed, which match well the chemical data of a K-feldspar from Deer et al. (2013). In comparison to EMPA data, light elements like Al and Na seem to be slightly overestimated by M4

Tornado quantification, but are still in an identifiable range of a K-feldspar. The workflow of the μ-EDXRF measurement and following analysis is displayed in the block-scheme of Fig. 2.

## 3    Results: Comparison between ENVI and MLA for plutonic rock thin sections

For comparison and verification of the μ-EDXRF-ENVI classification, three thin sections of plutonic rocks were analyzed and classified with ENVI and compared to MLA. The classification results and the mineral distribution maps are shown in Fig. 3,

the modal mineralogy of both methods is listed in Table 3.

In general, the mineral distribution maps of both classifications correspond well with the thin section photo. Both methods recognized the present minerals. Single grains can be identified in thin section and both mineral distribution maps. Minerals that sometimes are difficult to differentiate in thin section microscopy like quartz and plagioclase can be identified and separated due to the chemical information, in which the plagioclase contains Si as well as Al, Na and Ca. Texture and grain

structures even of complex intergrowth are recognizably well mapped. There can be found few differences in details, though: MLA is able to detect microstructures like micro-pertithic intergrowth in sample 424, due to the smaller beam diameter and the sampling depth limited to a few micrometers, whereas the data based on the M4 Tornado measurement integrates information of a 17 μm spot size. In the ENVI classification of sample 342 clinopyroxene grains surrounded by plagioclase have sometimes small rims of hornblende. This is due to the overlap of both minerals producing mixed signals which are

chemically similar to hornblende. The modal mineralogy differences of both classifications and the three samples are displayed in detail in Fig. 4. It shows the mentioned discrepancies for the feldspars. However, the regression coefficient $R^2$ of 0.98 shows a very good correlation between the modal mineralogy of both methods. An error matrix was calculated with ENVI after resizing (Nearest Neighbor) the MLA classification for 50% for technical reasons and image to image registration (Table 4). The error matrix shows fair overall accuracy of 76%. There seem to be three classes of accuracy: The first class with good

accuracy of about 80 % consists of K-feldspar, quartz, allanite and hornblende. The second class with fair to medium accuracy between 60-70 % consists of ilmenite, plagioclase and orthopyroxene. Minerals with low accuracy are clinopyroxene, which is mostly confused with hornblende, then magnetite, zircon and apatite, which are mostly unclassified in MLA.



For a detailed visual comparison of both classifications, a section of sample 342 is shown in Fig. 5. As mentioned before, small plagioclase veins cannot be identified with the M4 Tornado; it is also noticeable that some minerals like allanite, ilmenite or orthopyroxene have small unclassified (white) rims in the ENVI map, which is due to the mixed fluorescence signals coming from different depth in the M4 Tornado measurement. Separation of clinopyroxene and hornblende is difficult, because both

minerals are chemically similar. Main elements like Si, Ca and Fe are present in both minerals and other elements like Al, K or Ti have very low X-ray fluorescence intensities in the M4 Tornado measurement due to their low content or low atomic number. Nevertheless, the grain outlines in general are well comparable.

## 4    Discussion

The proposed method depends mainly on the correct mineral classification. The key is to create a comprehensive mineral

database that contains all present minerals and is able to distinguish minerals of similar spectral features. Having information about the geological system of the sample and the possible paragenesis will improve the classification and decrease the occurrence of unclassified areas. Since many minerals are parts of solid solution series e.g. plagioclase, pyroxene or biotite, the mineral database can consist of several endmembers of one mineral group in order to classify chemical changes within one solid solution series.

Isochemical minerals like rutile, anatase and brookite ($TiO_2$) are not distinguishable with an M4 Tornado measurement. Only with further information from other methods like Raman Spectroscopy or X-ray diffraction information about the crystal system could be obtained and used for the classification. Until then, identification of the mineral is based on the chemical information. But, the classification can be extended from mineral groups to mineral endmembers easily, when more detailed information is available. The rock classification should "grow with the science" (Carr and Hibbard, 1991).

A similar problem creates the range of detectable elements. Since the lightest detectable element is Na, minerals that contain lighter elements are not clearly identifiable. Apatite, for example, can be identified by the abundance of phosphorus, calcium and possibly chlorine. Distinguishing between flourapatite and hydroxylapatite is not possible with this method. Therefore, the group name apatite should be used. A special case is Fe. If there is a mineral containing Fe solely, there are several possibilities: The iron oxides magnetite and hematite would fit the chemical data since oxygen is not detectable. The presence

of Ti would indicate magnetite or titanomagnetite. Iron hydroxide and oxide-hydroxide fit the chemical data, too, as well as siderite (Fe carbonate). Little amounts of Ca, Mg or Mn would indicate the iron carbonate, but still several possibilities remain for Fe.

Furthermore, the resolution limits of each device have to be taken into account. Looking at the modal analysis, differences occur mostly from the estimation of quartz and plagioclase and the intimate intergrowth of feldspars. According to the MLA

measurement, sample 484 contains about 5% less plagioclase than what M4- classification determined. These differences result from the micro-perthitic intergrowth of the sample. The perthite shows small lamellae of plagioclase in the K-feldspar host. When these lamellae are smaller than the beam diameter of 17 µm, the pixel will be classified as alkali-feldspar with elevated



Na and Ca content, while a small fraction of plagioclase will be lost in the modal mineralogy. Similarly, differences can occur from other mineral combinations or overlaps in one pixel, which may result even in a different mineral classification or unclassified grain boundaries. Plagioclase overlapping with pyroxene can be chemically similar to hornblende and result into a small hornblende rim between plagioclase and pyroxene grains. Minerals with low classification accuracy like zircon,

magnetite and apatite are present mostly as small and single grains surrounded by e.g. plagioclase. This results in an unclassified rim due to overlapping signals of both minerals. Due to the small grain sizes, the number of unclassified border pixels form a relatively great proportion compared to the number of core pixels and, therefore, result in a very low classification accuracy.

Comparing the M4 classification procedure to MLA, some difficulties should be mentioned. With MLA, differentiation of

minerals with similar atomic number is difficult, when the measurement should involve all rock forming and accessory minerals from very low to very high atomic numbers and resulting grey levels. The grain separation in XBSE mode is based on the grey value of the BSE image. Minerals of similar atomic number will be eventually combined (e.g. plagioclase / quartz / muscovite or biotite / hornblende / pyroxene) and only one measurement in the center of the particle will be done. Contrast and brightness of the BSE images are adapted for the present mineral assemblage. If a good contrast for distinction of light

minerals like quartz and plagioclase is needed, heavier minerals like allanite or zircon appear white and adjacent minerals of equal grey value will not be separated. Therefore, it was necessary to measure the thin section of sample 484 in GXMap mode, which increased the measuring time from 12 to 40 h. Another source of error was found to be the frame overlapping. Some frames are shifted apart and the automatic particle joining had to be corrected manually several times.

The combination of µ-EDXRF and hyperspectral classification shows good applicability for heavy accessory minerals and

sulfide ores, since heavy elements are easy to detect due to their good fluorescence response. Even small grains of accessories can be detected, since the EDXRF-mapping provides spatially resolved data. If there is a sample area of 20 x 15 cm with just one gold grain large enough to detect, the element distribution map will show the gold grain. ENVI offers the possibility to locate such minor minerals of interest and provide the pixel coordinates of each grain of interest. Furthermore, the mineral distribution maps offer opportunities to perform image analysis, e.g. for calculation of geometric grain parameters (Nikonow

and Rammlmair, 2016).

An important factor to consider for scientist and laboratories are financial aspects, such as purchase and maintenance as well as easy handling and usability of the devices. In this case, the financial advantage is on the side of the µ-EDXRF. The acquisition costs for the µ-EDXRF including two detectors and the software (ENVI) is about 250.000 €, whereas the SEM including the MLA software may be around four times higher. To operate an SEM, high vacuum pumps and a nitrogen supply

are necessary as well as a skilled operator, whereas the µ-EDXRF can operate at atmospheric pressure or in low vacuum and is relatively easy to operate for both scientists and also students. Both devices should preferably be operated in an air-conditioned laboratory, however, in this work the µ-EDXRF was not in an air-conditioned room.

Taking into account the limits of µ-EDXRF, it is important to analyze the question or the problem the data should answer or solve. The big advantage is the little preparation and measuring time. It is possible to have chemical information of a large





sample area within hours after having taken the samples. When chemical analysis can take days to pulverize or digest the samples, EDXRF-mapping can give a good overview in a short time period, within a certain error limit, though. For detailed chemical analysis either the bulk sample has to be processed, or small areas or minerals have to be separated for the detailed analysis, which is very time-consuming. µ-EDXRF provides spatially resolved chemical data and therefore even small areas

of interest can be analyzed separately. For microscopy and petrographic analysis thin sections have to be prepared, which are limited in most cases to an area of a few cm². Microscopy is very helpful, but it can be a great advantage to be able to see more than a few thin sections, especially the area that is in between them. Since the EDXRF-maps are fast to obtain, depending on the size and chosen acquisition time, results can be obtained within minutes or several hours, respectively.

## 5    Conclusions

In this work we describe the multispectral classification of plutonic rock thin sections based on µ-EDXRF data. The SAM classification was shown to work well for primary, mostly unweathered plutonic rocks. Compared to MLA, the mineral classification results correspond well on thin sections. Problems arise due to the technical limits of the used µ-EDXRF instrument including resolution and not measurable elements, whereas a lot of valuable information of even larger samples than thin section size can be obtained faster with multispectral classification.

Working non-destructively and covering an area of 20 x 15 cm, this method is well suitable to obtain a sample overview with chemical, textural and mineralogical information and even geometric grain information.

Furthermore, it can be seen as the first step in a series of geoscientific analyses. It can help choosing the areas of interest for thin section preparation. Acquiring spatial and chemical information about the samples can decrease the number of necessary thin sections or the following chemical analyses, since the choice can be made more targeted and systematic. Overall, it is an

20 objective, repeatable and quantifiable way for modal mineralogy and petrographic image analysis.

*Competing interests.* The authors declare that they have no conflict of interest.

*Acknowledgements.* The results of this work are part of research that is funded by the German Federal Ministry of Education and Research (BMBF) within the project SecMinStratEl (grant no. 033R118B). The authors are thankful to Jeannet Meima for the many helpful comments on the manuscript, Katarzyna Krasniqi for parts of the mineral database, Dominic Göricke for
technical support with the SEM and Gerhard Heide from the Bergakademie TU Freiberg for the fruitful discussions.

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

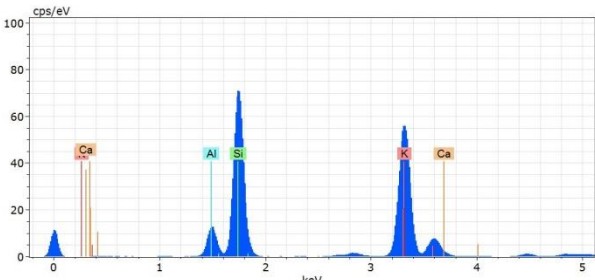

Fig. 1: µ-EDXRF spectrum of a K-feldspar with the main constituents Al (1.04 keV), Si (1,7 keV) and K (Kα: 3,3 keV, Kβ: 3,6 keV)

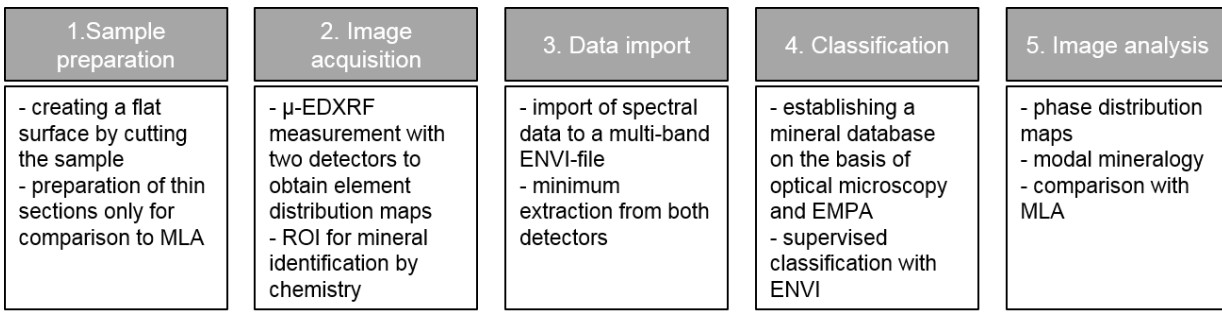

Fig. 2: Block-scheme for the workflow of the µ-EDXRF analysis



| Sample | Thin section photo | MLA Classification | M4 Classification |
|--------|--------------------|--------------------|-------------------|
| 424 | | | |
| 342 | | | |
| 484 | | | |

**Fig. 3: Thin section scan and classification results from Mineral Liberation Analyzer (MLA) and M4 Tornado - ENVI. The size of thin sections 424 and 484 is about 30x20 mm; thin section 342 is about 40 x 25 mm; the color key for the classified minerals is displayed in Fig. 3.**





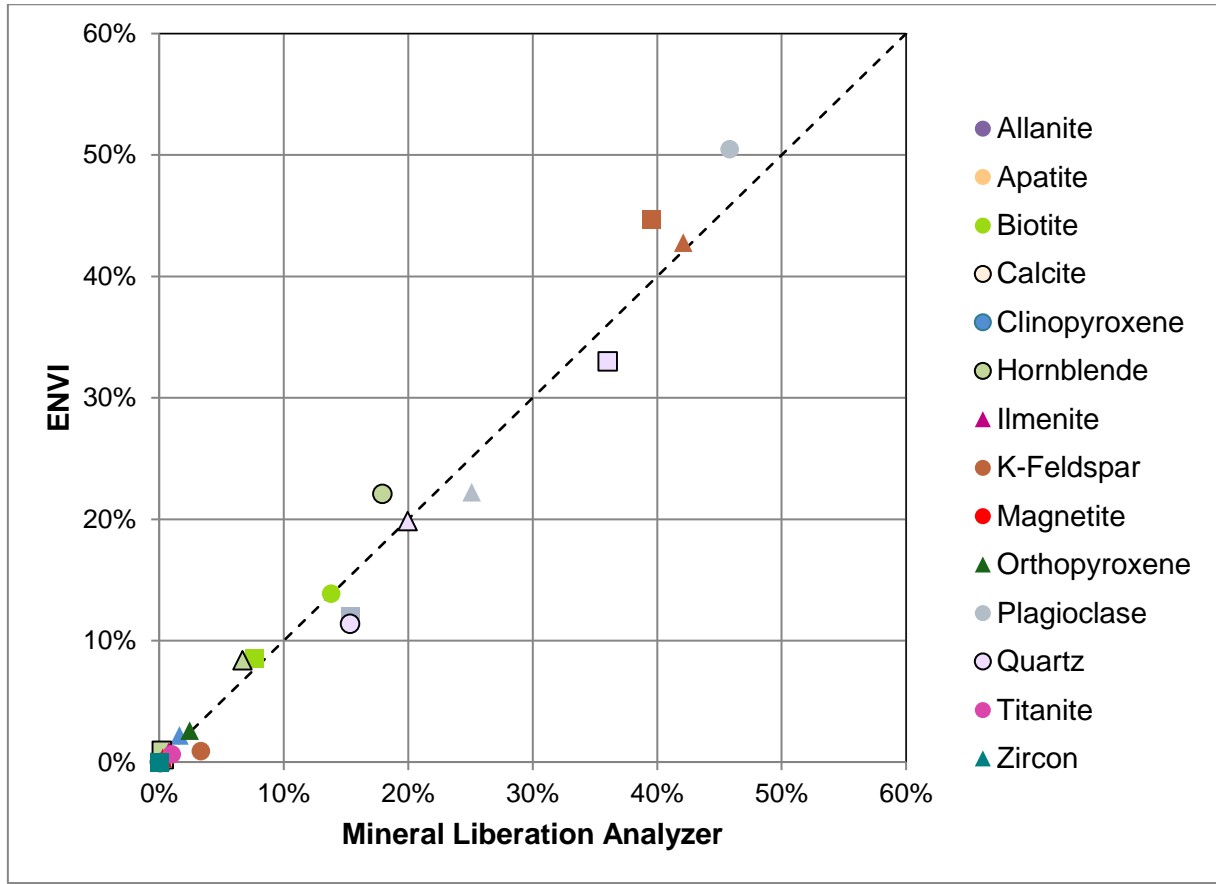

**Fig. 4: Modal mineralogy according to ENVI and Mineral Liberation Analyzer classification for samples 484 (circle symbols), 424 (square) and 342 (triangle) with data from Tab. 3.**





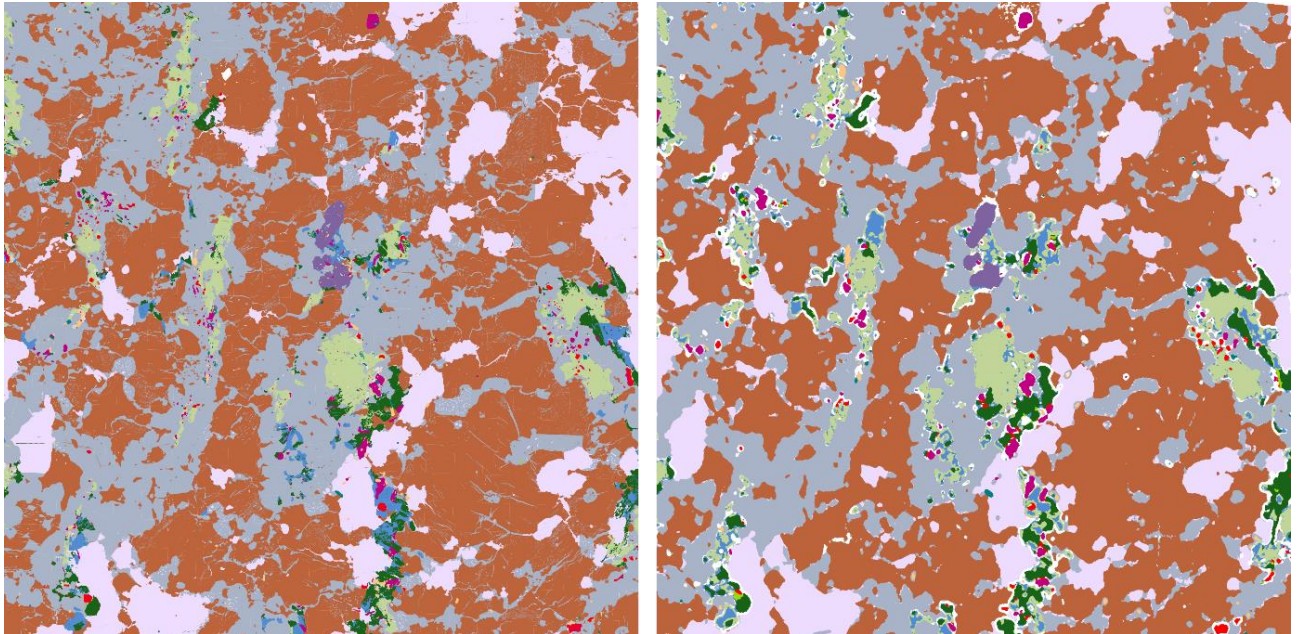

**Fig. 5: Detailed view of a section in the upper right corner of sample 342: Mineral distribution map from MLA (left) and ENVI (right). Each image width is about 15 mm. See Fig. 4 for the color key.**

**Table 1: Details of the M4 Tornado mapping and SEM + MLA measurement**

| Device | M4 Tornado | FEI Quanta 650 F<br>MLA XBSE | FEI Quanta 650 F<br>MLA GXMAP |
|---|---|---|---|
| Sample type | ca. 25 x 35 mm thin section | 28 x 38 mm thin section | 28 x 38 mm thin section |
| Exciting Energy | 50 kV, 600 µA | 25 kV, 222 µA | 25 kV, 222 µA |
| Step size | 12 µm | 1,5 µm (BSE) | 6 µm |
| Dwell time per spot | 2 ms | 7 ms per grain | 7 ms |
| Measurement time | 6 h | 9-12 h | 40 h |



**Table 2: M4 Tornado chemical analysis of a K-feldspar spectrum from Fig. 1, EMPA from three different spots on K-feldspar and literature value from Deer et al. (2013)**

|  | M4 Tornado | EMPA 55/1 | EMPA 56/1 | EMPA 64/1 | Literature |
|---|---|---|---|---|---|
| SiO$_2$ | 63.57 | 64.30 | 63.77 | 63.53 | 64.66 |
| TiO$_2$ | 0.40 |  |  |  | 0.00 |
| Al$_2$O$_3$ | 21.65 | 18.13 | 18.62 | 18.16 | 19.72 |
| Fe$_2$O$_3$ | 0.22 | 0.18 | 0 | 0 | 0.08 |
| MgO | 0.66 |  |  |  | 0.00 |
| CaO | 0.49 |  |  |  | 0.34 |
| Na$_2$O | 1.88 | 0.22 | 0.22 | 0.27 | 3.42 |
| K2O | 11.09 | 15.90 | 16.29 | 15.61 | 11.72 |
| Sum | 99.96 | 98.71 | 98.9 | 97.57 | 99.94 |

5   **Table 3: Modal mineralogy from the M4 classification in area-% (MLA indicates thin section analyzed by MLA in XBSE mode, GX indicates MLA in GXMAP mode, M4 indicates thin section analyzed by the M4 Tornado. <0.0 means that the mineral was detected in a quantity less than one decimal.**

|  | 424 MLA | 424 M4 | 342 GX | 342 M4 | 484 MLA | 484 M4 |
|---|---|---|---|---|---|---|
| Unclassified | 0.6 | <0.0 | <0.0 | <0.0 | 1.9 | <0.0 |
| Quartz | 36.0 | 33.0 | 20.0 | 19.9 | 15.3 | 11.4 |
| K-Feldspar | 39.6 | 44.7 | 42.1 | 42.3 | 3.3 | 0.9 |
| Plagioclase | 15.4 | 12.0 | 25.1 | 22.2 | 45.8 | 50.5 |
| Hornblende | 0.2 | 0.9 | 6.7 | 8.4 | 17.9 | 22.1 |
| Biotite | 7.6 | 8.6 | 0.1 | <0.0 | 13.8 | 13.9 |
| Calcite | 0.4 | 0.3 | ND | ND | 0.1 | <0.0 |
| Clinopyroxene | ND | ND | 1.6 | 2.2 | 0.1 | <0.0 |
| Orthopyroxene | <0.0 | <0.0 | 2.4 | 2.6 | ND | ND |
| Magnetite | <0.0 | <0.0 | 0.3 | 0.4 | 0.2 | <0.0 |
| Ilmenite | ND | ND | 0.8 | 0.9 | ND | ND |
| Titanite | ND | ND | ND | ND | 1.0 | 0.6 |
| Apatite | <0.0 | <0.0 | 0.3 | 0.2 | 0.5 | 0.3 |
| Zircon | 0.1 | <0.0 | 0.2 | 0.1 | <0.0 | <0.0 |
| Allanite | <0.0 | <0.0 | 0.4 | 0.4 | <0.0 | <0.0 |





**Table 4: Error Matrix in percent for sample 342 with the MLA as reference data in the columns and the μ-EDXRF in lines with mineral abbreviations as follows: Unclassified Uncl., Clinopyroxene Cpx, Hornblende Hbl, Allanite Aln, Magnetite Mag, Ilmenite Ilm, Quartz Qtz, K-Feldspar Afs, Zircon Zrn, Apatite Ap, Orthopyroxene Opx.**

| Class | Uncl. | Cpx | Hbl | Aln | Mag | Ilm | Qtz | Pl | Afs | Zrn | Ap | Opx | Total |
|-------|-------|-----|-----|-----|-----|-----|-----|----|-----|-----|----|-----|-------|
| Uncl. | 58,21 | 6,54 | 5,84 | 8,21 | 20,06 | 17,93 | 8,36 | 8,26 | 6,17 | 40,9 | 19,11 | 11,77 | 9,68 |
| Cpx | 0,42 | 38,18 | 11,24 | 3,08 | 7 | 0,91 | 0,52 | 1,71 | 0,12 | 8,7 | 5,76 | 6,96 | 2,11 |
| Hbl | 1,14 | 38,22 | 77,92 | 3,04 | 20,91 | 5,81 | 1,34 | 4,92 | 0,45 | 5,45 | 11,5 | 10,56 | 7,63 |
| Aln | 0,02 | 0,06 | 0,01 | 79,96 | 0,07 | 0,16 | 0,05 | 0,06 | 0,03 | 0,02 | 0,49 | 0,01 | 0,32 |
| Mag | 0,04 | 0,54 | 0,42 | 0,02 | 41,77 | 9,89 | 0,02 | 0,27 | 0,03 | 1,57 | 1,66 | 1,92 | 0,34 |
| Ilm | 0,12 | 0,56 | 0,45 | 1,57 | 0,6 | 63,96 | 0,1 | 0,71 | 0,08 | 7,22 | 6,9 | 0,51 | 0,78 |
| Qtz | 3,75 | 0,43 | 0,05 | 0,25 | 0,14 | 0,07 | 79,55 | 4,48 | 3,8 | 0,74 | 0,57 | 0,25 | 18,01 |
| Pl | 13,63 | 1,9 | 1,02 | 2,16 | 1,72 | 0,54 | 6,47 | 66,13 | 5,04 | 10,71 | 15,82 | 0,59 | 20,07 |
| Afs | 22,13 | 0,04 | 0,04 | 0,19 | 0,19 | 0,02 | 3,13 | 12,37 | 84,11 | 1,54 | 0,95 | 0,06 | 38,49 |
| Zrn | 0 | 0,04 | 0,01 | 0,26 | 0,02 | 0,02 | 0,02 | 0,07 | 0,03 | 20,21 | 0,44 | 0 | 0,06 |
| Ap | 0,04 | 0,31 | 0,38 | 0,51 | 0,01 | 0,03 | 0,03 | 0,25 | 0,04 | 1,19 | 22,54 | 0,01 | 0,18 |
| Opx | 0,5 | 13,17 | 2,61 | 0,74 | 7,53 | 0,67 | 0,4 | 0,76 | 0,12 | 1,77 | 14,27 | 67,36 | 2,33 |
| Total | 100 | 100 | 100 | 100 | 100 | 100 | 100 | 100 | 100 | 100 | 100 | 100 | 100 |