# Peer review of "Automated mineralogy based on energy dispersive X-ray fluorescence microscopy (µ-EDXRF) applied to plutonic rock thin sections in comparison to Mineral Liberation Analyser"

_Geoscientific Instrumentation, Methods and Data Systems, 2017_

## Referee Comment (RC1) · Anonymous Referee #1 · 2 Jul 2017

The manuscript is original and it has the potential to become a useful contribution; however, it is too long and difficult to follow, but can be easily shortened by removing the irrelevant data (and information; the authors should evaluate the meaning and need of each sentence and method applied in the manuscript). The sentences are very long and the English should be checked.

If there are some discrepancies in the minerals percent in every measurement method,

what is the suggested method for future data measurement?

Page 3, line 32: "a polarized microscope "instead of "a polarization microscope" Page 3, line 23: This sample was measured in GXMAP mode, where the whole sample was mapped with a distance of 6 $\mu$m. What do you mean? Figs.2 and 4: Image scale was not shown.

---

## Referee Comment (RC2) · Anonymous Referee #2 · 15 Aug 2017

1. Why use the algorithm, Spectral Angle Mapper, in classification? Please provide a bit explanation

2. I don't really understand Table 4, error matrix. Please explain more about 'error matrix'.

3. One of advances of the method u-EDXRF is to use less efforts (time) than other methods. My understanding is it uses larger step size according to Table 1. Please
confirm.

4. Fig.4 Clarify only marker colors of the legend on the figure are meaningful.

---

## Author Comment (AC1) · 23 Aug 2017

Dear referee,

we are thankful for your helpful comments and remarks on our manuscript and appreciate your time and effort for the review. Below you will find our responses to the comments and our changes to the manuscript.

[Figure]

1. The manuscript is original and it has the potential to become a useful contribution; however, it is too long and difficult to follow, but can be easily shortened by removing the irrelevant data (and information; the authors should evaluate the meaning and need of each sentence and method applied in the manuscript). The sentences are very long and the English should be checked.

Response: We have edited our manuscript and worked on the length and the language. In order to shorten the manuscript we omitted the paragraph in the introduction about development and history of automated mineralogy and left only the most important information and references regarding MLA. We also tried to clarify and shorten sentences where possible. Furthermore, we removed Fig. 4 (Modal mineralogy according to ENVI and MLA classification) and the corresponding remarks in the results section. Even though it shows a great correlation, the error matrix is more informative, since it shows differences and accuracy for each mineral class.

2. If there are some discrepancies in the minerals percent in every measurement method, what is the suggested method for future data measurement?

Response: As always in geosciences, the preferred method depends on the scope of the analysis and the necessary data resolution. We have addressed the options and opportunities of both methods in the last paragraphs of our discussion and conclusion and have added more detailed information and application suggestions for the relevant technique. Furthermore, it is not necessarily only a question of which method to use, but it can be a big advantage to use both methods, first $\mu$-EDXRF and then, if a higher resolution is needed, MLA. Our suggestions are to use $\mu$-EDXRF as a first step of a series of geoscientific analyses including thin section microscopy, MLA or other geochemical analyses, since it is fast and non-destructive. That means, having the $\mu$-EDXRF information with an intact sample remaining, the decision, where to take thin sections, or which part of the sample should be analyzed geochemically can be done more systematic and more targeted.

3. 3.1. Page 3, line 32: "a polarized microscope "instead of "a polarization microscope"

Response: Corrected to "polarized light microscope"

3.2. Page 3, line 23: This sample was measured in GXMAP mode, where the whole sample was mapped with a distance of 6 _m. What do you mean?

Response: The information on the two mapping modes and their function are now explained in a clearer structure and with more details as follows in chapter 2.3 and in table 1: "For this work, two modes of MLA were applied: (1) samples 424 and 342 were measured in the XBSE mode, where grains are classified and separated according to their grey level in the BSE image. Then each separated grain is measured in the center with the X-ray detectors and classified chemically using a predefined mineral database. For the sample 484 the XBSE mode was not suitable, since the grey values of hornblende and biotite were too similar for a correct grain separation. Therefore, this sample was measured in (2) GXMAP mode. In this mode, grains are not separated by their grey values, but the whole sample was continuously mapped with an EDX-analysis every 6 $\mu$m. The details of the SEM image acquisition are listed in Table 1. A detailed description of the functionality of MLA and the measuring modes can be found elsewhere (Dobbe et al., 2009; Fandrich et al., 2007; Gu, 2003)."

3.3. Figs.2 and 4: Image scale was not shown.

Response: The sample sizes of Figs 3 and 5 were given in the image captions. For better visibility, scales have been integrated into both images.
* * *
| | Thin section photo | MLA Classification | M4 Classification |
|---|---|---|---|
| 424 | | | |
| 342 | | | |
| 484 | | | |

**Fig. 1.** New Fig. 3: Thin section scan and classification results from Mineral Liberation Analyzer (MLA) and M4 Tornado - ENVI.

[Figure]

Allanite
Apatite
Biotite
Calcite
Clinopyroxene
Hornblende
Ilmenite
K-Feldspar
Magnetite
Orthopyroxene
Plagioclase
Quartz
Titanite
Unclassified
Zircon

**Fig. 2.** New Fig. 4: Detailed view of a section in the upper right corner of sample 342: Mineral distribution map from MLA (left) and ENVI (right).

[Figure]

---

## Author Comment (AC2) · 23 Aug 2017

Dear referee,

thank you for your time and effort reviewing our manuscript. We have considered your helpful remarks and included the necessary additional information in the manuscript as follows:

1. Why use the algorithm, Spectral Angle Mapper, in classification? Please provide a bit explanation

Response: The following information and literature reference has been added to chapter 2.4: "The classification algorithm allocates a mineral name to each pixel in the element distribution map according to a prior defined database of mineral spectra (endmember collection). It calculates the spectral similarity of two spectra, which is described by the angle between the vectors of both spectra. The angle of the spectral similarity can have values from 0 to $\pi/2$ in radians (Masoumi et al., 2017). The vectors are in an n-dimensional space, where n is the number of bands (here: element lines). SAM was developed for classification of hyperspectral images and is most widely applied in context with mineralogical classification (Van der Meer and De Jong, 2003). Girouard et al. (2004)."

2. I don't really understand Table 4, error matrix. Please explain more about 'error matrix'.

Response: The following explanation regarding the error matrix was added to the introduction: "Assessment of 2D classification data has been applied and discussed widely among remote sensing scientists. In most cases, hyperspectral images are evaluated by comparison to reference images (Foody, 2002), which are supposed to have true classification values (ground truth images) e.g. through manual control. Each pixel of the reference image is compared to the new classification and the numbers of pixels assigned to each class are entered into an error matrix (or confusion matrix); the reference pixels are listed in columns and the new data in rows. The central diagonal represents the pixels that were assigned to the correct class, all others have been assigned to a different class. The classification's overall accuracy can be calculated by dividing the sum of the correctly classified pixels (central diagonal) by the total pixel number (Congalton, 1991)."

3. One of advances of the method u-EDXRF is to use less efforts (time) than other

methods. My understanding is it uses larger step size according to Table 1. Please confirm.

Response: The effort for a measurement is a combination of sample preparation, measurement preparation and the actual measurement. The $\mu$-EDXRF is non-destructive and needs almost no sample preparation, since it only needs a flat sample surface. In our work, polished thin sections were used only for a good comparison to MLA. Also, the operational effort of a $\mu$-EDXRF is only a fraction of the effort to operate a SEM+MLA (laboratory, equipment, skilled operator, polishing, carbon coating and measurement).

With MLA we applied two different measurement modes: (1) XBSE uses grey levels to differentiate single grains and performs only one analysis per identified grain. That means there is no regular measurement grid, which can save a lot of measuring time, if minerals have enough contrast to be distinguished. Because the grey level differentiation did not work on sample 484 due to similar average atomic numbers of minerals, we applied (2) GXMAP, which is a continuous mapping with a given step size. This increased the measurement time to about 40 h, which is almost 7 times the $\mu$-EDXRF measurement.

While the SEM uses an electron beam at 25 kV, $\mu$-EDXRF uses an x-ray beam at 50 kV for excitation. The high excitation voltage allows the $\mu$-EDXRF to measure K-lines of elements that exceed the SEM range. Well known examples are Zr or Mo with K-lines above 15 keV and overlapping L-Lines with the K-lines of P and S, which can be easily distinguished by $\mu$-EDXRF.

4. Fig.4 Clarify only marker colors of the legend on the figure are meaningful.

Response: We have added the color key and legend of all phases to the figure, even if this small section does not show some minerals like titanite. This is necessary because Fig.3 uses the same color code and it would consume a lot of space to include a color key in Fig.3.

[Figure]

**Fig. 1.** New Fig. 4; Detailed view of a section in the upper right corner of sample 342: Mineral distribution map from MLA (left) and ENVI (right).

---

## Referee Comment (RC3) · Anonymous Referee #2 · 25 Aug 2017

I am happy to see all the concerns/questions that have been answered. I am satisfied with the response. No more questions and concerns and would agree to publish.

---

## Editor Comment (EC1) · L.V. Eppelbaum (Editor) · 31 Aug 2017

As a geophysicist with rich field experience, I must note that the automated express mineralogy determinatioins are very useful for optimization of geological-geophysical investigations and prompt decision making.

[Figure]

https://doi.org/10.5194/gi-2017-33, 2017.